# Professional quality of life in animal research personnel is linked to retention & job satisfaction: A mixed-methods cross-sectional survey on compassion fatigue in the USA

**Lauren Young**[1,2], **Fabienne Ferrara**[3], **Lisa Kelly**[4], **Tara Martin**[5], **Sally Thompson-Iritani**[6], **Megan R. LaFollette**[2]*

1 Department of Integrative Biology, University of Guelph, Guelph, Ontario, Canada, 2 The 3Rs Collaborative, Denver, Colorado, United States of America, 3 Consulting and Training in Animal Research, Berlin, Germany, 4 University of Georgia, Athens, Georgia, United States of America, 5 Refinement and Enrichment Advancements Laboratory, Unit for Laboratory Animal Medicine, University of Michigan, Ann Arbor, Michigan, United States of America, 6 Department of Environmental and Occupational Health Sciences, Office of Research, University of Washington, Seattle, WA, United States of America

* meglafollette@na3rsc.org

## Abstract

### Background

Working with research animals can be both rewarding and challenging. The rewarding part of the work is associated with understanding the necessity for animal research to improve the health of humans and animals and the knowledge that one can provide care and compassion for the animals. Challenges with animal research include witnessing stress/pain in animals necessitated by scientific requirements, end of study euthanasia, and societal stigmatization about animal research. These challenges could be compounded with more general workplace stresses, in turn, impacting job retention and satisfaction. However, these factors have yet to be formally evaluated. Therefore, the purpose of this survey was to comprehensively evaluate professional quality of life's correlation with key workplace metrics.

### Methods

Six institutions were recruited to participate in a longitudinal intervention trial on compassion fatigue resiliency. This manuscript reports key baseline metrics from this survey. A cross-sectional mixed methods survey was developed to evaluate professional quality of life, job satisfaction, retention, and factors influencing compassion fatigue resiliency. Quantitative data were analyzed via general linear models and qualitative data were analyzed by theme.

### Results

Baseline data was collected from 198 participants. Personnel who reported higher compassion satisfaction also reported higher retention and job satisfaction. Conversely, personnel who reported higher burnout also reported lower job satisfaction. In response to open-ended questions, participants said their compassion fatigue was impacted by institutional

**Data Availability Statement:** All relevant data supporting the findings of this study are available

within the article and its supplementary information files. The dataset used in this study is available in the University of Michigan's Deep Blue Data repository, accessible at https://doi.org/10.7302/cdpa-qp43.

**Funding:** MRL received funding for this project from the National Institute for Occupational Safety and Health (NIOSH) under Federal Training Grant T42OH008433. https://deohs.washington.edu/nwcohs/research/pilot-funding The funders had no role in study design, data collection and analysis, decision to publish, or preparation of the manuscript.

**Competing interests:** All authors are employed by the institution noted in their affiliation except for Lauren Young who was a student during data collection and initial writing and then transitioned to a full-time role with the 3Rs Collaborative during manuscript revision. All authors are also members of the 3Rs Collaborative's Compassion Fatigue Initiative. Elizabeth Nunamaker and Sally Thompson-Iritani also sat on the board of directors and leadership team of the 3Rs Collaborative during publication. This does not alter our adherence to PLOS ONE policies on sharing data and materials.

culture (70% of participants), animal research (58%), general mental health (41%), and specific compassion fatigue support (24%).

## Conclusions

In conclusion, these results show that professional quality of life is related to important operational metrics of job satisfaction and retention. Furthermore, compassion fatigue is impacted by factors beyond working with research animals, including institutional culture and general mental health support. Overall, this project provides rationale and insight for institutional support of compassion fatigue resiliency.

## Introduction

Professionals that work with research animals can experience both distinct rewards and challenges due to the unique nature of their work. They contribute to critical scientific research that benefits the lives of people and animals such as through developing treatments for diseases and contributing to scientific breakthroughs. They often work in dynamic professional roles that help safeguard and enhance the lives of research animals themselves [1,2]. Both factors can provide great meaning for these professionals. Conversely, to accomplish these key scientific aims, it may be necessary to approve, view, or cause pain and/or distress to research animals as part of the experimental paradigm. This can lead to a sticky moral situation that we term the "Caring-Harming Paradox" and at the end of a study, although some animals may be rehomed or adopted, many are euthanized for reasons such as post-mortem scientific examinations or due to biohazard concerns [3,4]. Additionally, these professionals can experience negative social stigma for working with research animals which can be exacerbated by misrepresentation of animal research by some individuals or organizations [1,5]. Despite finding their work meaningful, these personnel may not feel as valued as other care workers [6].

Beyond the unique factors related to working with research animals, these professionals are also subject to general workplace factors that can influence burnout. As can occur in any field (although especially caring professions) staff may be asked to work long hours, feel unappreciated by other sectors of their institution, and face interpersonal conflicts with colleagues [6,7]. They may not fully understand their own or other key job responsibilities in the organization. Good work-life balances practices may not be promoted well in the face of organizational pressures. Furthermore, personal life and mental health challenges from outside of work can impact experiences at work [8–10].

Both the unique factors of working in animal research and typical workplace stressors can lead to decreased professional quality of life. Professional quality of life in caring professions can be segmented into compassion satisfaction (e.g., feeling good about their caring role) and compassion fatigue (e.g., feeling challenged by their caring role). Compassion fatigue is often described as being comprised of secondary traumatic stress (which is like post-traumatic stress disorder but caused by exposure to the stress of others) and burnout [11,12], considered by many as a manifestation of chronic psychological strain [13].

Research thus far has shown that compassion fatigue in animal research personnel is associated with several factors. One of the strongest findings is that personnel with less social support experience higher compassion fatigue [3,14,15] and evident by both quantitative and qualitative studies, seems to be a moderating factor for the development of psychological strain in the laboratory [13]. Animal-related findings include exposure to higher animal stress/pain [3,14],

less enrichment frequency/diversity [3], physical euthanasia methods [3,13], close animal relationships [7], more euthanasia [16] less control over performing euthanasia [3,7,16] and internal conflicts between animal welfare and scientific needs [17–19]. General mental and physical factors are also associated with compassion fatigue such as poor mental or physical health as well as lower emotional stability, openness, and extraversion [7]. Finally, poor relationships with superiors and a lack of training and resources for coping with compassion fatigue have been identified as negative factors [7].

Although previous literature has begun to establish associations between professional quality of life and potential risk factors, a few gaps remain. Although a few studies have connected professional quality of life to general workplace factors such as staffing and workplace relationships [7], no studies have yet connected professional quality of life to key factors of interest to workplace management such as retention and job satisfaction. Job satisfaction can be described as the feelings or attitudes an individual has in relation to their job and workplace [10]. Based on the literature, job satisfaction seems to be the most important factor for working motivation, performances and job retention [10,20,21], which is important to maintain an adequately qualified workforce and reduce turnover [22]. Thus far, most research has primarily used close-ended questions and limited free response, and there has been no extensive qualitative research on the subject to our knowledge.

Considering the gaps in previous research on workplace stress in research personnel, our objective in this survey was to explore associations between reported professional quality of life (i.e., compassion fatigue and satisfaction) and important workplace metrics (e.g., retention and satisfaction) and explore factors that individuals report as impacting their compassion fatigue resiliency. We hypothesized that higher compassion fatigue and lower compassion satisfaction would be associated with lower workplace satisfaction and retention. We also hypothesized that participants would cite key factors related to mental health support and working with research animals as impacting their compassion fatigue. With this knowledge, we hope to provide additional rationale and guidance for interventions promoting compassion fatigue resiliency for animal research personnel.

## Materials and methods

All procedures and waived signed consent protocols were approved by University of Michigan's Human Research Protection Program Institutional Review Board (IRB), protocol # HUM00207730. Participants provided waived signed consent via a yes or no question on the online survey platform after reading an informed consent document. No IACUC approval was sought as there were no interactions between the researchers or animals during this study.

### Participants and procedures

Institutions were recruited to participate in this study via direct email or verbal communication with previously known contacts, presentations by members of the 3Rs Collaborative, and a request via the 3Rs Collaborative newsletter. Institutional inclusion criteria included being located in the USA or Canada, either not currently having a compassion fatigue resiliency program or having a newly established program, and be willing to work closely with the 3Rs Collaborative for recruitment and implementation of a program. In this manuscript, we present baseline data from a 2-year longitudinal study investigating the effectiveness of implementing institutional compassion fatigue resiliency programs.

Ultimately six institutions met inclusion criteria and were able to participate in the study. The institutions included one academic institution, three research institutes, and two large pharmaceutical companies. Institutions could choose to allow participation of only animal

care staff or all related research personnel (e.g., researchers, IACUC members, etc.), based on the intended recipients of their individual planned compassion fatigue resiliency program. Each institution contained a possible sample size between 7 and 429 individuals. Altogether they represented approximately 723 eligible participants.

At each institution, one to three local contacts were identified to coordinate participation. These contacts were typically directors, managers, or supervisors with authority to coordinate compassion fatigue resiliency activities. These contacts recruited participants between Feb 11 and March 22, 2022, via three email contacts and one physical flyer. To compensate them for their time, participants were entered into a random drawing for a $25 visa gift card. Inclusion criteria for participants were being over the age of 18 and currently working at one of the included institutions; there were no exclusion criteria.

Following reading an informed consent document with the assurance that responses would be kept confidential, including from supervisors, participants confirmed documentation of waived signed consent. They then completed an online questionnaire estimated to take an average of 10 minutes via Qualtrics. Participants were informed that they could skip any question that made them feel uncomfortable. Although 3 of the authors had access to email addresses that could identify individual participants during data collection, during the data analysis phase the responses were de-identified with a participant and institutional code to ensure they were kept anonymous and confidential. Potentially identifiable information was only accessible to core research team members.

## Measures

The 3Rs Collaborative's Compassion Fatigue Resiliency committee created a mixed-method cross-sectional survey based on a review of the literature and consultations with experts in survey methodology and laboratory animal science. When possible, the survey contained validated survey instruments (e.g., professional quality of life scale [1]), but when such items did not exist similar survey scales were modified for purpose (e.g., modified nurse retention index, [2]) or created new for purpose. When new scales were created, they were reviewed by our team, piloted, and revised as necessary.

Overall, participants were asked 78 to 85 questions. The additional questions were asked only to personnel that worked with research animals in a hands-on role to attempt to determine if retention was unique for these types of roles. Questions were subdivided into 5 subsections as described below. All survey text and scoring can be found in **S1 Table**.

## Demographics & work factors

After gaining documentation of waived signed consent, participants were asked their age for inclusion and their email to allow linking of responses across yearly surveys. Additional work and demographic factors were then asked including working role, years of work in the field, sex, average hours of work in a week, and highest education. Participants were also asked if they currently worked hands-on with research animals to allow for segmentation. Finally, they were asked to report the degree of stress/pain that most animals in their care experience based off the official United States Depart of Agriculture pain and distress categories for laboratory animal research [3] as this has previously been shown to impact professional quality of life [4].

**Professional quality of life knowledge and experiences.** Participants were asked direct questions about their own self-reported compassion fatigue knowledge and experiences. They were first asked in close-ended questions of their familiarity with the definition of compassion fatigue, effective strategies to combat compassion fatigue, their own implementation of strategies to combat compassion fatigue, and whether they had experienced compassion fatigue in

the past. Then, participants were asked to rate their level of compassion fatigue on a descriptive one to five scale. Finally, participants were asked two open-ended questions about what makes compassion fatigue worse or better for them personally.

Participants were then asked to complete the 30-question professional quality of life scale (PROQOL) to determine compassion fatigue (comprised of burnout and secondary traumatic stress) and compassion satisfaction [1]. The PROQOL is a widely used instrument to determine the positive and negative aspects of caring for others.

**Job satisfaction & retention.**   Participants then completed scales to assess job satisfaction and retention. For satisfaction, participants were asked to complete the seven item Brief Index of Affective Job Satisfaction Scale which includes 3 distractor questions [5]. It asks participants to evaluate how much they agreed or disagreed with four statements about their current job. The scale ranges from 4 being very low job satisfaction and 20 being high job satisfaction. Then participants were asked to complete a modified nurse retention index [2] where "nursing" was replaced with "Research animals". The original MNRI has six questions including four positively worded questions and two negatively worded items. The scored scale ranges from 6 being very low planned retention to 48 being very high retention. Additionally, participants who worked hands-on with research animals were then asked a modified nurse retention index substituting "hands-on with research animal" where appropriate.

**Institutional program.**   To evaluate implementation of future institutional programs, participants were then asked two final questions about the program. First, participants were asked what they thought would be the most beneficial aspect of an institutional compassion fatigue resiliency program. Then, participants were asked which program components they planned to participate in.

### Data analysis

**Quantitative analysis.**   Quantitative data were analyzed with descriptive statistics and general linear models. Continuous data are presented as mean and standard deviation (SD). Counts are presented as $n$ and percent (%). Any duplicate responses were identified via matching email addresses; the most complete or recent response was retained. Only participants that answered questions through rating their level of compassion fatigue were included. For use in general linear models, categorical data with less than 20 responses were collapsed into larger categories. Additionally, summary scales were calculated according to instructions for each individual scale.

General linear mixed models were run to test associations between professional quality of life and both retention and job satisfaction. The dependent variables were retention and job satisfaction. The independent variables included compassion satisfaction, burnout, secondary traumatic stress, work factors (animal stress/pain, hands-on work, role, years of work, and hours per week), and demographics (highest education, age, sex). Institution was included as a random blocking factor. Significance level was set at $p < 0.05$. Results are presented as mean +- standard deviation. Effect sizes are reported using Cohen's $f^2$, where when $f^2 > = 0.02, 0.15$, and $0.35$ indicating a small, medium, and large effect size, respectively [23].

This representative analysis was used: Retention = Compassion Satisfaction + Burnout + Secondary Traumatic Stress + Perceived Stress Scale + Animal Stress/Pain + Hands-On Work with Animals + Role + Years + Hours per Week + Highest Education + Sex + Institution

**Qualitative analysis.**   Open-ended questions were assessed using inductive, bottom-up, content analysis to derive themes from all respondent answers. This process resulted in the formation of a coding manual used to identify common themes. The complete manual is found in **S2 Table.** The same manual was created and used to code all open-ended qualitative questions. Microsoft Excel was used for manual creation and thematic analysis.

Coding manual creation involved an iterative and collaborative process with multiple steps and researchers. The goal was to extract all general themes from the open-ended survey question that respondents identified to create one coding manual. This began with one researcher (LEY) reading through all written responses and noting the themes within each question. These themes were compiled into a master list, and in collaboration with a second researcher (MRL), themes were discussed to look for similarities, differences, relationships between themes and overlap with the Culture of Care terminology. A preliminary manual was created by grouping the identified themes. The data was subsequently re-read and coded based on the preliminary manual. This process was repeated to refine the coding manual and ensure all themes in the data were represented in the final coding manual.

The coding process itself involved breaking down responses to identify themes to subsequently code. Each response was broken down into grammatical clauses and based on the theme in the coding manual it described, each clause was given a code. There was no limit to the number of codes a single response was given, and every clause was coded. For example, one respondent stated their compassion fatigue is made worse by "having to euthanize many animals on a single day". This was coded as the theme Research Animals, and the subtheme 'euthanasia'. When possible, responses were coded with a subtheme, to increase specificity. When a response did not clearly fall into a subtheme, they were coded only with the main theme. Non-comprehensive responses were coded as ambiguous.

Coding was performed by LEY and inter-rater reliability was assessed by having an additional individual, who was not involved in the manual creation process, code a random 20% of the data.

We calculated the prevalence of each theme by taking the number of participants whose response was coded with a particular theme/subtheme, divided by the total participants that responded to the survey. The formula used ensured each theme was only counted once per respondent, even if mentioned more than once across all qualitative questions.

## Results

### Quantitative

Of the approximately 723 potentially eligible participants, a total of 302 survey responses were started, but only 198 individuals were included in the survey as they gave responses at least through the first block of questions through rating their level of compassion fatigue resiliency. From the total response pool, included participants results in a response rate of 27%. The number of participants per institution ranged from 9 to 54.

Complete demographics are reported in **Table 1**. Participants were primarily animal caretakers (30%) and researchers (30%) although many other roles were represented. Most worked hands-on with animals (84%) and most worked with animals experiencing level 2 stress/pain (minor stress or pain of short duration, 51%). The majority of participants worked at research institutes (55%) or pharmaceutical organizations (39%). Most participants either had the highest degree of a bachelors (43%) or their veterinary or graduate degree (34%). Participants were age of 38 ± 12, primarily female (72%), worked 40 hours a week (57%), and just over half had worked in the field for 10 years or more (51%).

**Professional quality of life knowledge and experiences.**   Research animal personnel reported about their compassion fatigue knowledge and experiences (**Fig 1**). Most participants (87%) agreed that they were familiar with the definition and components of compassion fatigue. Most also agreed that they had experienced compassion fatigue in the past (70%). A little more than half agreed that they understood effective strategies for combatting compassion

**Table 1. Demographic and work information for survey participants (N = 198).**

| Role | N | % of Total |
|---|---|---|
| Animal Caretaker | 60 | 30% |
| Researcher | 59 | 30% |
| Manager | 18 | 9% |
| Research Technician | 16 | 8% |
| Veterinarian | 10 | 5% |
| Veterinary Technician | 10 | 5% |
| Other | 15 | 8% |
| **Highest Education** | | |
| Graduate or Veterinary Degree | 66 | 34% |
| Bachelor's Degree | 84 | 43% |
| Associate Degree | 21 | 11% |
| High School Diploma | 26 | 13% |
| **Institution Type** | | |
| Research Institute | 109 | 55% |
| Pharmaceutical Organization | 78 | 39% |
| Academic | 11 | 6% |
| **Sex** | | |
| Female | 143 | 72% |
| Male | 50 | 25% |
| Transmale or Transfemale | 2 | 1% |
| Prefer Not to Answer | 3 | 2% |
| **Animal Stress/Pain** | | |
| Little or no discomfort of stress | 35 | 18% |
| Minor stress or pain of a short duration | 92 | 51% |
| Moderate stress or pain of a short duration | 55 | 29% |
| Procedures which cause severe pain near, at, or above the pain tolerance threshold of unanesthetized conscious animals | 4 | 2% |
| **Hours per Week (Categorized) (n = 197)** | | |
| < 40 | 21 | 11% |
| 40 | 112 | 57% |
| 41–49 | 22 | 11% |
| 50+ | 42 | 21% |
| **Years Working (Categorized) (n = 194)** | | |
| < 10 | 92 | 46% |
| 10 to 19 | 60 | 30% |
| 20+ | 42 | 21% |

fatigue (58%). However, less than half (44%) agreed that they had implemented strategies to combat compassion fatigue.

Participants reported their compassion fatigue via a descriptive scale and the professional quality of life scale which are presented in categorized form in **Table 2**. Based on a descriptive scale, most participants indicated they felt occasional burnout or stress but not compassion fatigue (64%), although a considerable portion felt they had symptoms of compassion fatigue (29%). Based on the PROQOL cutoff scores, no participants had low compassion satisfaction, high burnout, or high secondary traumatic stress. Based on continuous data (**Table 3**), on average, participants experienced moderate compassion satisfaction, burnout, and secondary

**Personnel generally were familiar with the definition/components of compassion fatigue and agreed they had experienced compassion fatigue in the past**

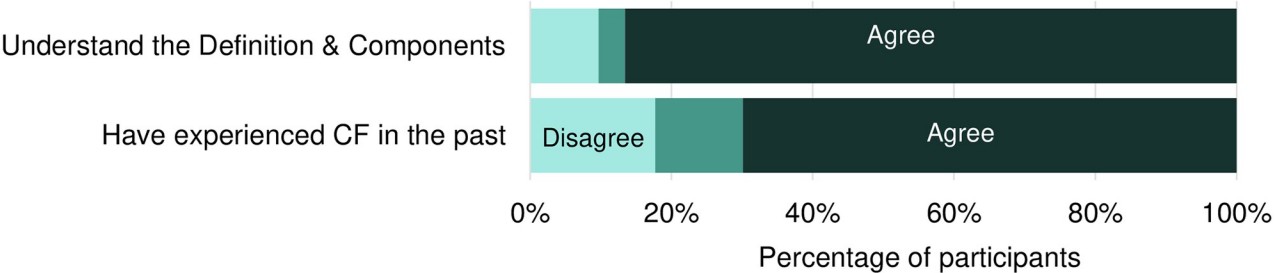

**A little more than half agreed they understood strategies for combatting compassion fatigue, but did not agree they had implemented them.**

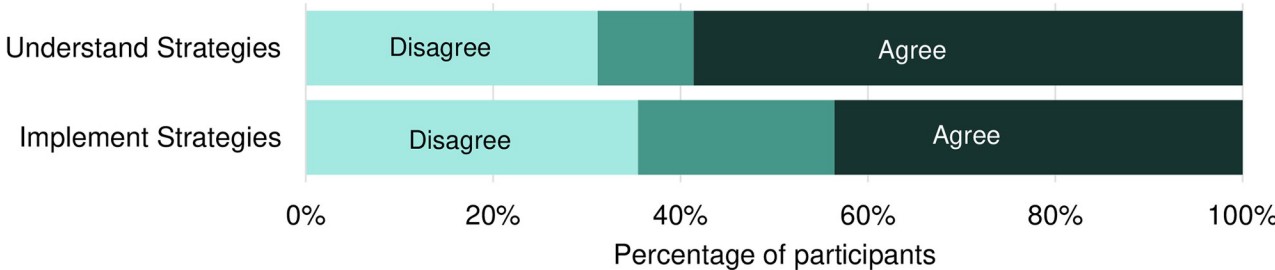

**Fig 1. Animal research personnel's experiences & understanding of compassion fatigue.** Animal research personnel (N = 198) from 6 institutions across the USA answered questions about their understanding on compassion fatigue, whether they had experienced it, and whether they both understood/implemented strategies to combat compassion fatigue.

traumatic stress. On average, participants agreed with statements that they were satisfied with their job (**Table 3**). Finally, participants reported higher planned retention to continue with their research animal career than to continue working hands-on with research animals (**Table 3**).

Participants (n = 166) anticipated participating in the following components of a compassion fatigue program in decreasing order: webinar on overview of CFR (60%), webinar on culture of care (54%), webinar on mindfulness (52%), webinar on communication & trust in the workshop (51%), enrichment activities (50%), accessing independent activities (45%), accessing mindfulness or gratitude materials (42%), accessing reading materials on compassion fatigue (40%), in person activities (39%), poster viewing (36%), participating in group activities (36%), webinar on meaning making (30%), and memorial activities (26%).

**Associations with job satisfaction, & retention.** In this survey, job satisfaction and retention were found to be associated with several factors (**Table 4**). Research animal personnel who reported higher job satisfaction indicated less burnout ($p < 0.0001$, $f^2 = 0.13$) with a small effect size and higher compassion satisfaction ($p < 0.0001$, $f^2 = 0.23$) with medium effect size. Research personnel who reported higher general retention reported higher compassion satisfaction ($p < 0.0001$, $f^2 = 0.16$) with medium effect size. Research personnel who worked in hands-on roles with research animals who reported higher retention indicated compassion

**Table 2. Descriptive compassion fatigue and professional quality of life.**

| Descriptive Compassion Fatigue (n = 195) | N | % of Total |
|---|---|---|
| I enjoy my work. I have no symptoms of compassion fatigue | 15 | 8% |
| Occasionally I feel burned out or feel some stress from my work with animals, but I don't really feel like I have compassion fatigue | 125 | 64% |
| I am definitely burning out and have symptoms of compassion fatigue from my work with animals. | 44 | 23% |
| The symptoms of compassion fatigue that I'm experiencing won't go away | 9 | 5% |
| I feel so much compassion fatigue and often wonder if I can go on. I am at the point where I may need some changes or may need to seek some sort of help. | 2 | 1% |
| **Compassion Satisfaction (n = 187)** | | |
| Low (22 or less) | 0 | 0% |
| Moderate (23–41) | 122 | 65% |
| High (42 or more) | 65 | 35% |
| **Burnout (n = 188)** | | |
| Low (22 or less) | 86 | 46% |
| Moderate (23–41) | 102 | 54% |
| High (42 or more) | 0 | 0% |
| **Secondary Traumatic Stress (n = 187)** | | |
| Low (22 or less) | 103 | 55% |
| Moderate (23–41) | 84 | 45% |
| High (42 or more) | 0 | 0% |

satisfaction ($p < 0.0001$, $f^2 = 0.11$) with small effect size and reported more average hours working per week ($p < 0.0184$, $f^2 = 0.04$) with a small effect size.

**Qualitative results.** A total of 85% (n = 167) participants responded to at least one open-ended, qualitative question. Participants responded to three questions: what makes your compassion fatigue worse (n = 160, 81%), what makes your compassion fatigue better (n = 156, 79%) and what would be most beneficial about a compassion resiliency program (n = 106, 54%). Theme prevalence is summarized across all three questions. Each subtheme's name and response frequency are noted in paratheses. A detailed summary of qualitative results can be found in **Fig 2** and **S2 Table.**

**Theme 1: The culture of my institution contributes to my compassion fatigue.** Nearly three-quarters of participants (n = 118, 71%) indicated that something related to institutional culture or organization, unrelated to animal research, impacted their compassion fatigue. Specifically, participants mentioned work-life balance, staff interactions, feeling valued, general organization, training, or pay. One response that captures a number of these themes' states, "our constant lack of adequate and reliable staff and the constantly growing list of things we as

**Table 3. Descriptive statistics for professional quality of life, general retention, hands-on retention, and job satisfaction.**

| | n | Mean | Standard Deviation |
|---|---|---|---|
| Compassion Satisfaction | 187 | 38.70 | 6.08 |
| Burnout | 188 | 23.43 | 5.84 |
| Secondary Traumatic Stress | 187 | 22.05 | 5.98 |
| General Retention Index | 185 | 36.41 | 11.65 |
| Hands-on Retention Index | 155 | 24.54 | 12.44 |
| Job Satisfaction | 183 | 15.96 | 3.17 |

**Table 4. Associations with job retention and satisfaction with research animal personnel.**

| Independent Variables | Retention (General) | Retention (Hands-On) | Job Satisfaction |
|---|---|---|---|
| **Professional Quality of Life** | | | |
| Compassion Satisfaction | ^^(+) $F_{1,171}$ = 22.11, p < 0.0001 | ^(+) $F_{1,145}$ = 15.32, p = 0.0001 | ^^(+) $F_{1,169}$ = 28.21, p < 0.0001 |
| Burnout | $F_{1,171}$ = 2.11, p = 0.1488 | $F_{1,145}$ = 2.53, p = 0.1144 | ^(-) $F_{1,169}$ = 23.01, p < 0.0001 |
| Secondary Traumatic Stress | $F_{1,171}$ = 0.02, p = 0.8847 | $F_{1,145}$ = 0.23, p = 0.6360 | $F_{1,169}$ = 2.8, p = 0.0961 |
| **Work Factors** | | | |
| Animal Stress/Pain | $F_{1,171}$ = 1.90, p = 0.1700 | $F_{1,145}$ = 1.00, p = 0.3185 | $F_{1,169}$ = 2.99, p = 0.0861 |
| Hands-on Work | $F_{1,171}$ = 1.58, p = 0.2103 | - | $F_{1,169}$ = 0.05, p = 0.8154 |
| Institution | $F_{5,171}$ = 0.56, p = 0.7260 | $F_{5,145}$ = 0.71, p = 0.6193 | $F_{5,169}$ = 0.80, p = 0.5528 |
| **Demographics** | | | |
| Hours per Week | $F_{1,171}$ = 3.16, p = 0.0775 | ^(+) $F_{1,145}$ = 5.70, p = 0.0184 | $F_{1,169}$ = 1.99, p = 0.1609 |
| Years in Field | $F_{1,171}$ = 3.87, p = 0.0511 | $F_{1,145}$ = 2.93, p = 0.0892 | $F_{1,169}$ = 0.83, p = 0.3639 |
| Role | $F_{4,171}$ = 1.59, p = 0.1810 | $F_{4,145}$ = 1.81, p = 0.1310 | $F_{4,169}$ = 0.51, p = 0.7311 |
| Age | $F_{1,171}$ = 1.42, p = 0.2356 | $F_{1,145}$ = 0.17, p = 0.6848 | $F_{1,169}$ = 0.29, p = 0.5938 |
| Sex | $F_{2,171}$ = 0.75, p = 0.4751 | $F_{2,145}$ = 0.75, p = 0.4724 | $F_{2,169}$ = 0.15, p = 0.8607 |
| Highest Education | $F_{2,171}$ = 0.74, p = 0.4813 | $F_{2,145}$ = 1.98, p = 0.1456 | $F_{2,169}$ = 0.51, p = 0.6029 |

The associations from three general linear models of research animal personnel's job retention and satisfaction. Job retention was evaluated both for working with research animals generally (n = 172) and in hands-on role (n = 146) by a modified nurse retention scale. Job satisfaction was measured via the affective job satisfaction scale (n = 170). Participants were asked to complete the professional quality of life questionnaire and about work factors (animal stress/pain, whether they worked hands-on, and their institution). They were also asked the hours they worked per week, years working in the field, job role, age, sex, and highest education. F = F-statistic. (+) the independent variable has a positive association with the dependent variable. (-) the independent variable has a negative association with the dependent variable. Bold indicates a significant effect. ^indicates a small effect size; ^^indicates medium effect size.

a department and I as a supervisor are behind on. Everyone is stressed, but when people snap at others, it doesn't help the situation and makes me want to not assist other areas."

Most often participants mentioned that maintaining a work-life balance, or lack thereof, is instrumental in their experience with compassion fatigue (subtheme = *work-life balance*, 44%). As one participant states, "lowering the expectations for scientists and allow[ing] them to create a healthy life-work balance" would be beneficial for compassion resiliency. Another emphasized the importance of "Vacations, mental health days, and coworkers who are willing to do some of the more stressful tasks when you are feeling burnt out." Generally, participants indicated that it was important to spend time away from the workplace to cope with stress and burnout.

About a third of participants indicated that interactions with other staff members, whether positive or negative, can impact compassion fatigue (subtheme = *interactions with staff*, 30%). One participant stated that a "lack of understanding from other colleagues or managers of the workload/tasks I have been assigned" makes compassion fatigue worse. Additionally, a large proportion of these responses specifically mentioned something related to feeling valued (subtheme = *feeling undervalued by staff*, 17%). Feeling undervalued or underappreciated at work makes compassion fatigue worse, while recognition, especially by organizational leads, makes compassion fatigue better. As one participant states, their compassion fatigue is relieved by "receiving recognition, understanding, and help from higher-ups in the department (manager and veterinarians)."

About a fifth of respondents mention general institutional factors, as opposed to the specific factors represented by other subthemes (subthemes = *general*, 20%). For example, one participant states that "those who are not involved from an animal use perspective making decisions

**Overall half of participants reported that their compassion fatigue is impacted by institutional culture & animal-research related factors.**

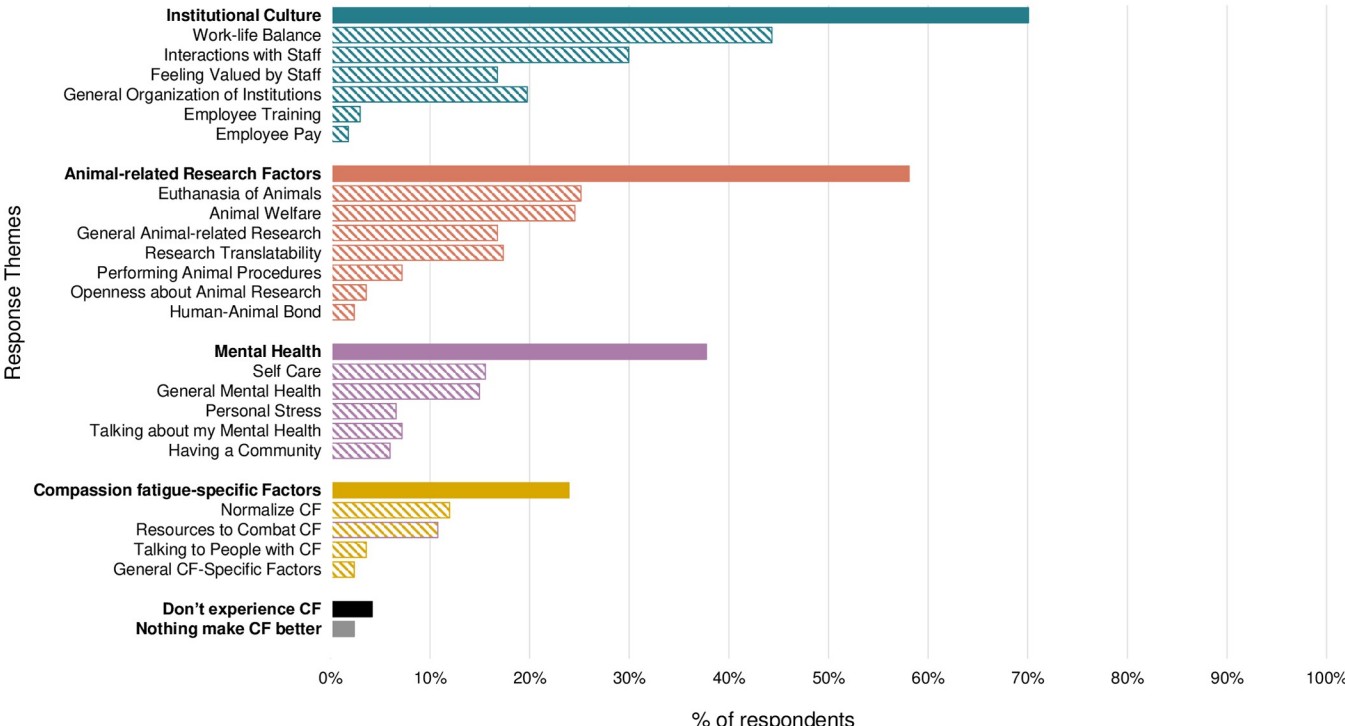

**Fig 2. Institutional, research and compassion fatigue-specific factors that impact compassion fatigue in research animal personnel.** The percentage of research animal personnel (n = 167) whose responses included each of the themes (solid bars) and subthemes (hashed lines) created in qualitative coding of all responses. Each color represents a different thematic category.

for animal staff" impacts their compassion fatigue. Others mention, "switching from my ex-vivo role and helping with in-vivo" and "the sense of knowing that tasks are completed".

Although not a particularly common response, we note that a small proportion of respondents mentioned pay (subtheme = *employee pay*, 2%) or training (subtheme = *employee training*, 4%) as impactful to their compassion fatigue. Participants stated more pay and more training are needed to reduce compassion fatigue. For example, one participant stated, "I would like to see changes in the structure of operations. The issues that we run into usually stem from lack of training. . .".

**Theme 2: Working with research animals can be challenging or rewarding.** Over half of participants (n = 97, 58%) mentioned animal research-related factors that are inherent to their work as contributing to their compassion fatigue. Specifically, participants mentioned performing euthanasia, procedures, changes in animal welfare, research translatability, openness about research or the human-animal bond.

A quarter of participants indicated that "having to euthanize animals" is a key contributor to their compassion fatigue (subtheme = *euthanasia of animals*, 25%). One participant stated, "I sometimes experience this now when working with the mice at my new job. For me, it's hard to bring them in alive, put them under and then decapitate them. It feels like it goes against everything I have done/ stood for the last few years; saving lives instead of taking them." Other participants indicated that performing procedures on animals can be challenging (subtheme = *performing animal procedures*, 7%). Examples include, "long, repeated procedures" and "large animals being dissected in the necroscopy room". For some individuals,

performing these types of procedures can contribute to compassion fatigue just as much as performing euthanasia.

A quarter of participants stated that witnessing animals with reduced welfare (e.g. in pain or suffering) impacts compassion fatigue (subtheme = *animal welfare*, 25%). As one participant stated, their compassion fatigue is worse "when a level of long-term pain or distress is necessary to meet the scientific requirements of a study". In particular, participants indicated it was difficult to handle events described in their words as "repeated", "high-intensity" or "unexpected". Conversely, participants indicated that reducing the frequency of these negative events would make compassion fatigue better.

About a fifth of participants mentioned that knowing and thinking about the translatability, "the big picture", and the "greater good" of animal research impacted compassion fatigue (subtheme = *research translatability*, 17%). These statements generally indicated that these thoughts made compassion fatigue better. As one participant stated, "[I] hope that my work with animals may one day produce a beneficial therapeutic that helps people manage disease and improve their lives." Some participants indicate that learning about the specific research aims of their own lab is helpful, stating, "I try to remember the important work being accomplished by our researchers. I talk to the researchers about their studies so I can put the animals and the procedures into context."

Another fifth of respondents mention general factors about working with research animals that impact compassion fatigue (subtheme = *general*, 16%). Some examples include "other people denying that animals feel pain" and "anything with the animals honestly".

A small number of participants also mention that openness, or lack thereof, about their work with animals can impact compassion fatigue (subtheme = *openness about animal research*, 4%). As one participant stated, "feeling so responsible for the animals in my care and working so hard to make sure they have the best lives possible but still receiving public backlash/lack of understanding".

Of note, a few participants state specific factors like "naming the animals", "learning their behaviours" or "playing with them" contribute to compassion resiliency (subtheme = *human-animal bond*, 3%). As one participant stated, "short studies are easier to feel compassion fatigue since the animals are not around as long and there isn't much of a bond between animals and handlers."

**Theme 3: My general mental health impacts my work life.** General mental health factors were commonly mentioned by participants when asked about compassion fatigue. A total of 38% of respondents (n = 64) made statements related to self-care, work stress, personal stress, talking about their mental health, or having a community.

Some participants mentioned using self-care practices (subtheme = *self care*, 17%) to help with compassion fatigue. Participants mentioned practices such as, "spending time with friends", "exercising", "thinking of happy thoughts", "funny movies, trying to take a walk outside, watching funny videos of kids and pets online" and "adequate sleep."

A portion of participants stated that their general mental health, or general stress levels were impacted (subtheme = *general*, 15%). Respondents make statements such as, "Heightened stress levels for prolonged periods" and "volume of stressful situations." One participant specifically states, "if your mental health is not being taken care of outside of work then you will be even more affected at work and by the work that you do", which emphasizes the importance of good mental health.

Respondents mentioned that talking to someone about their feelings, either a close colleague or a professional, can help their mental health, improving compassion resiliency (subtheme = *talking about mental health*, 8%). As one participant stated, it would be beneficial to "give employees an outlet to understand their feelings, talk about them and have them help makes changes as needed."

Additional respondents indicated that when their personal lives were stressful then that made their compassion fatigue worse (subtheme = *personal stress*, 7%). For example, participants mentioned factors such as "pets or children [they] are responsible for" or "upsetting personal experiences". One participant clearly described how work and personal stress can overlap stating, "deadlines, stresses when large experiments are coming up with heavy expectations. This can be made worse if my personal life is also heavy with stress or pain."

Some participants mentioned that group events and creating a sense of community are important (subtheme = *having a community*, 6%). For example, one participant stated that it would be beneficial to have "a supportive community with regular gatherings".

**Theme 4: I need compassion-fatigue specific help.** Compassion fatigue specific factors and resources were mentioned by approximately a quarter of participants (n = 40, 24%). They indicated the importance of promoting awareness of compassion fatigue, resources to combat it, and talking to others with compassion fatigue.

Participants discussed that there can be lack of knowledge or even stigma around compassion fatigue–and that normalizing compassion fatigue would be beneficial (subtheme = *normalize CF*, 12%). As one participant states, "just acknowledging [compassion fatigue] is a huge first step." Participants expressed a desire for compassion fatigue awareness to be widespread from upper management, to staff from other departments, and even the public. For example, one respondent stated it would be beneficial to "educating research staff on compassion fatigue and what care staff go through in a day."

Other participants highlighted the need for more resources to combat or prevent compassion fatigue (subtheme = *resources to combat CF*, 11%). As a few participants state, "recognizing the signs would be beneficial" and "teaching employees coping mechanisms for combatting compassion fatigue before it gets serious". One participant emphasizes that each individual may experience compassion fatigue in a unique manner, stating "there are many different factors that contribute to compassion fatigue, and not everyone will experience it in the same way. . .with a matter as serious as compassion fatigue, it is crucial to make sure that every individual does know how to have their needs with regards to receiving support met."

A few participants mentioned that hearing real-world stories from those who have experienced compassion fatigue before is helpful (subtheme = *talking to people with CF*, 4%). As one participant states, "discussion/talks by real people sharing their experiences–it makes me feel less alone when I see others who feel the same way I do." Some of these responses linked sharing experiences with normalizing compassion fatigue and therefore were coded in both categories. For example, as one participant states, "more discussion and sharing of personal experiences for the purposes of acknowledging this is a common and shared experience."

A couple of participants mentioned general compassion fatigue-specific factors that did not align with any of our other subthemes (subtheme = *general*, 2%). One participant says, "having an organization/workplace that is very aware of compassion fatigue" makes their compassion fatigue better.

Of note, 4% of participants state they do not experience compassion fatigue, and 2% state that nothing impacts their compassion fatigue levels.

**Themes by role.** *Post hoc*, we further investigated the percentage of respondents in each role who mentioned each theme (**Fig 3**). We investigated this for animal caretakers and researchers; the two participant groups with a sufficient sample size to make accurate conclusions based on subsequent results. The majority of animal caretakers state that their institutional culture (n = 37, 74%) and working with research animals (n = 27, 54%) contributes to their compassion fatigue. In comparison, researchers more often mention research animals (n = 31, 66%), and discuss institutional culture slightly less (n = 28, 59%). Both roles equally mention the impact their general mental health has on their compassion fatigue. Finally,

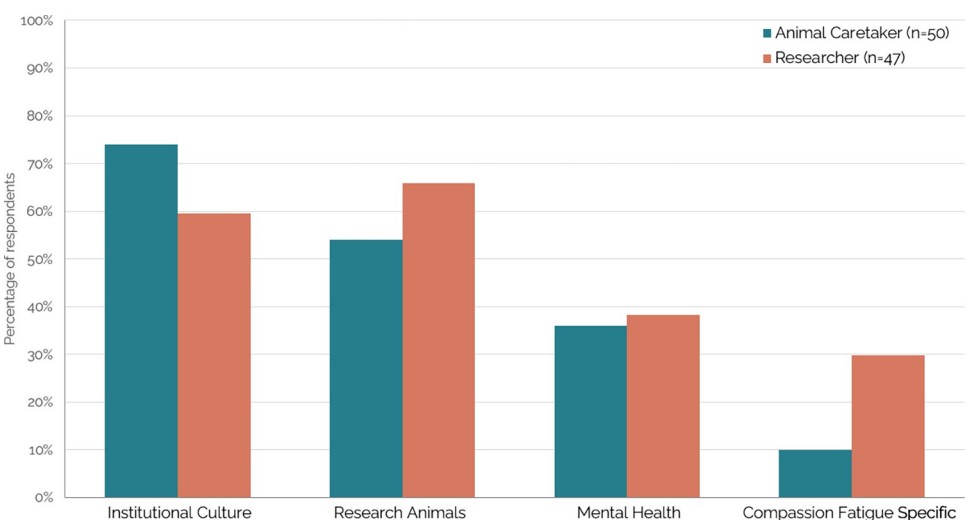

**Fig 3. Animal caretakers mention institutional culture, whereas researchers mention research animals and compassion fatigue specific factors more often when asked about their compassion fatigue.** The percentage of animal caretakers (n = 50) and researchers (n = 47) whose responses included each of the themes created in qualitative coding in at least one of their responses. Each color represents a different personnel role.

researchers mention compassion fatigue-specific factors (n = 14, 29%) more often than animal caretakers (n = 5, 10%).

## Discussion

To our knowledge, this is the first cross-sectional survey to find that professional quality of life in research animal personnel, including compassion satisfaction and burnout, is associated with retention and job satisfaction in animal research. Additionally, it is one of the first large qualitative surveys of compassion fatigue which found personnel reported the importance of institutional culture, factors unique to animal research, general mental health factors, and a desire for targeted resources. We successfully surveyed 198 individuals across 6 independent institutions and 9 unique sites.

### Professional quality of life is linked to job satisfaction and retention

In this survey, compassion satisfaction was positively associated with both job satisfaction and retention with a medium effect size. That is, personnel who reported higher levels of compassion satisfaction also reported being more likely to be satisfied with, and stay in, their current jobs. Furthermore, this association was found regardless of whether individuals currently worked hands-on with research animals and when personnel who were currently working in a hands-on role were asked about staying in a hands-on role.

The link between compassion satisfaction, job satisfaction, and retention may be unsurprising, as compassion satisfaction could be considered a particular subtype of job satisfaction. Additionally, our findings mirror previous work, that faculty in higher education that reported higher compassion fatigue also reported less intended retention [24]. And in turn, past research has found that less burnout and higher job satisfaction is associated with higher retention of nurses [8,21,22,25]. This survey provides important evidence of this linkage in the research animal personnel field across roles using previously validated scales.

In this survey, burnout (as assessed as a key component of compassion fatigue through the professional quality of life scale) was negatively associated with job satisfaction. That is,

personnel who reported higher levels of burnout also reported lower levels of job satisfaction. In the nursing field, a high quality of communication and information exchange between management and employees was negatively correlated with burnout [8] supporting the idea that as job satisfaction increases, burnout decreases.

Taken together, these findings provide a good rationale for institutions to promote professional quality of life. Not only are decreasing burnout and increasing compassion satisfaction good for personnel individually, both may also help prevent expensive and time-consuming employee turnover.

## It is more than just research animals: Culture and mental health matter

In this survey, we asked participants open-ended questions about what makes their compassion fatigue better and worse, as well as what they believed would be beneficial in a compassion fatigue resiliency program. Through analyzing the responses of 167 personnel across roles (including both caretakers and researchers), we were able to gain new insights into what factors may matter most in promoting professional quality of life in this field.

Significantly, the most common response given by participants was that factors related to institutional culture were important in their professional quality of life. In this way, the people working with research animals are just like any other employee in any other workplace. They want to feel supported in their home life, have positive interactions with other staff that lead them to feeling valued, and receive adequate training and pay. In fact, previous research has found increased compassion fatigue is linked to understaffing, feeling valued, long working hours, poor relationships/communication with superiors, and a generally poor work environment [3,6,7,16]. Similarly, in a systematic review of emergency healthcare workers many typical work- and job-related factors were found to lead to burnout [8]. These factors include interpersonal conflict and lack of supervisory support [26,27], quality of staff communication [28], adequate staffing [29], and workload [27].

Similarly, in our qualitative responses, participants often stated that their general mental health impacts their compassion fatigue. Again, just as in any other workplace, personnel discussed the importance of self-care and community, the ability to talk with others about their mental health, and stressors in their personal lives. Similarly, three previous studies have found worse compassion fatigue in animal research personnel who reported less social support or increased loneliness [3,14,15], while social support [13] and promoting mental well-being is suggested as an important component to building compassion-fatigue resiliency in an animal facility [30]. Additionally, previous work has found that poor personal mental health contributes to worse compassion fatigue, while talking to others is a good coping strategy [7]. Finally, a recent investigation found that high levels of mental well-being are positively associated with compassion satisfaction [14].

Together, these two findings highlight the ways that working in animal research has similarities to other jobs. These findings could be considered quite positive as they indicate that general strategies and workplace wellness programs designed for general institutional staff could also benefit animal research personnel's professional quality of life, and, in turn, retention. Furthermore, these findings point to potentially solvable problems, despite some of the inherent challenges of working with research animals.

## Working with research animals is uniquely rewarding and challenging

Despite our findings of the similar concerns of animal research personnel to other workplaces, there are still unique aspects of animal research that impact compassion fatigue. On the one hand, our participants state that it can be rewarding to contribute scientific advances, bond

with research animals and promote research animal welfare. These results are supported by previous findings that higher enrichment levels are associated with less burnout [3] and involvement and insight into research are relevant to psychological strain [13]. It is therefore possible that ensuring and communicating translational studies, promoting good relationships with animals, and generally promoting animal welfare could increase compassion satisfaction. In turn, retention may also be increased.

Conversely, our participants discussed the challenges of performing euthanasia, stressful procedures, witnessing an animal with reduced welfare, and societal stigma contributing to worsened compassion fatigue. Again, these findings are supported by previous research that worse compassion fatigue is associated with personnel reporting increased animal stress and pain, less control of euthanasia, physical methods, and greater euthanasia distress [3,6,15]. These factors align with research suggesting that unique aspects of the research animal environment can contribute to compassion fatigue [31,32]. Interestingly, our quantitative analysis didn't reveal a direct association between either animal stress/pain or hands-on animal work with retention or job satisfaction.

*Post hoc*, we were able to investigate similarities and differences between two roles distinct to the animal research setting: animal caretakers and researchers. Although with a small sample size and unable to investigate this for all the research animal personnel surveyed, this suggests different roles within the animal research setting may be impacted uniquely. Further research is needed to adequately investigate this topic.

## Compassion-fatigue specific resources are beneficial

Finally, our survey results suggest that the provision of specific resources may help alleviate compassion fatigue. Participants discussed the importance of compassion fatigue-specific support which included normalizing and reducing stigma with compassion fatigue, talking to others with compassion fatigue and having targeted resources and strategies to combat it. Indeed, previous research has found that providing training and resources for coping with compassion fatigue is linked to improved compassion fatigue [6,7].

Investigations of healthcare professionals during the COVID-19 pandemic found that social stigma was associated with increased compassion fatigue and decreased compassion satisfaction [33]. This is especially pertinent to research animal professionals experiencing compassion fatigue, as there is the negative stigma generally associated with animal care workers combined with the ongoing stigma surrounding mental health [34,35]. Lastly, numerous studies support the notion that talking to others is helpful through talk therapy [36] and specifically talking to others with a shared experience through group therapy [37,38].

Specific resources for compassion fatigue may include educating staff on what compassion fatigue is, recognizing its signs, and outlining the steps for prevention or mitigation. Additionally, programs may attempt to normalize and decrease the stigma surrounding compassion fatigue. To our knowledge, two institutions have published about their compassion fatigue programs: University of Washington [39] and Ohio Status University [16]. Additional institutions, such as the University of Michigan, also provide online resources detailing their programs. These programs could be used as models of institutional programs. Providing specific resources to promote professional quality of life has the potential to increase job satisfaction and retention in research animal personnel.

## Limitations & generalizability

This survey includes key limitations that are important to acknowledge. As this was a cross-sectional survey, it is not possible to determine the causation, if any, of determined

associations. That is, it's possible that rather than poor professional quality of life causing decreased job retention that instead individuals who do not plan to stay in their job experience poor professional quality of life due to that choice. A randomized empirical intervention trial would be necessary to determine any direction of causation. However, this survey still provides further rationale for the importance of institutional compassion fatigue resiliency programs for animal research personnel and provides guidance for future research.

Additionally, this survey was limited as we might have failed to capture information from personnel currently experiencing compassion fatigue and by design would have missed information from those who have already left the field. Individuals with high levels of compassion fatigue may have been less likely to respond to this survey due to workplace withdrawal and decreased motivation. Individuals that already left the field would not have been reached due to distribution being through current workplace emails and networks. If anything, these limitations may cause our findings to be stronger than what was found here. It is also important to note that although we surveyed research personnel at 6 institutions, this was not a representative sample of all personnel working in animal research across the United States. Therefore, our results may not be generalizable beyond this particular sample. Despite these limitations, our findings still provide insight into the lives of those currently working in the research field in the United States.

## Conclusions

In conclusion, these results show that research animal personnel professional quality of life is linked to two critical workplace factors: job satisfaction and retention. Furthermore, research animal personnel in the United States are impacted not only by the work they do with research animals and whether they have been provided compassion fatigue specific resources, but also by their general institutional culture and mental health support. These results suggest that institutions that focus on improving compassion satisfaction and decreasing compassion fatigue could improve employee satisfaction and retention. To accomplish these aims, institutions may benefit from improving workplace culture, improving specific animal research factors, providing general mental health support, and providing compassion fatigue specific resources. Ultimately these results provide insight and rationale for improving the professional lives of a critical sector of our society that conducts animal research.

## Supporting information

**S1 Table. Survey text and coding scheme.** The question text and answers shown to participants as well as the corresponding variable name, scale, and coded value of each answer.
(XLSX)

**S2 Table. Qualitative coding manual, results, and descriptions.** The name of each thematic code, a generalized participant response created by the researchers, subtheme response % and n, main theme response % and n, description of each category, key phrases, and representative quotes.
(XLSX)

## Acknowledgments

We gratefully acknowledge the institutions and research animal personnel who took the time to participate in and promote this survey. We also thank the 3Rs Collaborative's staff, volunteers, and members for making this research possible. We appreciate all research animal

personnel who have worked to promote institutional compassion fatigue resiliency. Finally, we thank all research animals currently being used in science.

The authors would first like to thank the institutions who chose to participate in this pilot study, especially for the individual champions who helped coordinate distribution of the survey. We also wish to thank all the research personnel who took the time to participate in this survey and provide useful insight, also acknowledging those who champion compassion fatigue resiliency efforts. We wish to thank that research animals used in research worldwide. Lastly, we would like to thank the 3Rs Collaborative staff, volunteers, and sponsors for making this research possible, and the participation from the entire compassion fatigue resiliency initiative on this project.

## Author Contributions

**Conceptualization:** Fabienne Ferrara, Tara Martin, Sally Thompson-Iritani, Megan R. LaFollette.

**Data curation:** Lauren Young, Megan R. LaFollette.

**Formal analysis:** Lauren Young, Megan R. LaFollette.

**Funding acquisition:** Sally Thompson-Iritani, Megan R. LaFollette.

**Investigation:** Tara Martin, Megan R. LaFollette.

**Methodology:** Lauren Young, Megan R. LaFollette.

**Project administration:** Tara Martin, Megan R. LaFollette.

**Resources:** Tara Martin, Megan R. LaFollette.

**Software:** Tara Martin.

**Supervision:** Tara Martin, Sally Thompson-Iritani, Megan R. LaFollette.

**Validation:** Lisa Kelly.

**Visualization:** Megan R. LaFollette.

**Writing – original draft:** Lauren Young, Fabienne Ferrara, Sally Thompson-Iritani, Megan R. LaFollette.

**Writing – review & editing:** Lauren Young, Fabienne Ferrara, Lisa Kelly, Tara Martin, Sally Thompson-Iritani, Megan R. LaFollette.

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
