## [Decision Letter · Decision Letter 0]

25 Oct 2023

PONE-D-23-17551Professional quality of life in animal research personnel is linked to retention & job satisfaction: A mixed-methods cross-sectional survey on compassion fatigue in the USAPLOS ONE

Dear Dr. LaFollette,

Thank you for submitting your manuscript to PLOS ONE. After careful consideration, we feel that it has merit but does not fully meet PLOS ONE’s publication criteria as it currently stands. Therefore, we invite you to submit a revised version of the manuscript that addresses the points raised during the review process.

Please calculate the effect size as requested by Reviewer 2.==============================

We look forward to receiving your revised manuscript.

Kind regards,

Rosemary Frey

Academic Editor

PLOS ONE

“All authors are employed by the institution noted in their affiliation except for Lauren Young who is a student. All authors are also members of the 3Rs Collaborative’s Compassion Fatigue Initiative. Elizabeth Nunamaker and Sally Thompson-Iritani also sat on the board of directors and leadership team of the 3Rs Collaborative during publication.”

4. Please remove your figures from within your manuscript file, leaving only the individual TIFF/EPS image files, uploaded separately. These will be automatically included in the reviewers’ PDF.

Reviewers' comments:

Reviewer's Responses to Questions

**Comments to the Author**

1. Is the manuscript technically sound, and do the data support the conclusions?

Reviewer #1: Yes

Reviewer #2: Yes

2. Has the statistical analysis been performed appropriately and rigorously? 

Reviewer #1: Yes

Reviewer #2: Yes

3. Have the authors made all data underlying the findings in their manuscript fully available?

Reviewer #1: Yes

Reviewer #2: Yes

4. Is the manuscript presented in an intelligible fashion and written in standard English?

Reviewer #1: Yes

Reviewer #2: Yes

5. Review Comments to the Author

Reviewer #1: Dear Authors,

Congratulations on your work. It is an interesting manuscript that continues with your line of work and that contributes new data to a field that has more consideration.

Regarding statistics, I would propose to calculate the size effect, and make correlations to determine the influence of each variable studied.

There is a typo in table 1, "veterinary technician" 910 instead of 10.

Best regards,

Reviewer #2: The manuscript by Young et al uses a mixed methods cross sectional survey to explore the quality of life of animal research personnel. Of particular interest is the finding that compassion fatigue is impacted most by institutional culture. On this point, it could be valuable for the authors to divide the respondents by level of seniority of the employee on the institutional culture metric.

As mentioned by the authors, this study cannot account for those who have left their role. However, the inclusion of such data (for example, the turnover rate) of specific job categories, could be insightful. Whether their current institution is their 1st or 5th, and the reason for their departure from previous roles could also provide good info. From my experience, there is often a very high rate of departure and hiring in animal care roles.

Line 274 - Table 1 under veterinary technician typo? I believe the authors meant to put n of 10, not 910.

Papers of potential interest to the authors for this study.

https://www.nature.com/articles/s41684-021-00852-6

https://labanimres.biomedcentral.com/articles/10.1186/s42826-022-00129-0

https://www.frontiersin.org/articles/10.3389/fvets.2020.573106/full

https://onlinelibrary.wiley.com/doi/full/10.1111/brv.13020

Overall, a good study.

6. PLOS authors have the option to publish the peer review history of their article (what does this mean?). If published, this will include your full peer review and any attached files.

Reviewer #1: No

Reviewer #2: No

---

## [Author Response · Author response to Decision Letter 0]

6 Dec 2023

November 15, 2023 

Dear Dr. Rosemary Frey 

Thank you for inviting us to submit a revised version of our manuscript. We appreciate the detailed and helpful feedback that both reviewers provided. We believe their contributions have strengthened our manuscript. We have completed all revisions and address each comment individually below. The reviewer's comments are bolded and our responses are non-bolded. 

We would also like to add the following requested statement to the Competing Interests section of our manuscript:

“This does not alter our adherence to PLOS ONE policies on sharing data and materials.”

We have also added the following three references to our reference list, based on helpful suggestions from Reviewer #2:

32. Murray J, Bauer C, Vilminot N, Turner PV. Strengthening workplace well-being in research animal facilities. Frontiers in Veterinary Science. 2020;7. doi:10.3389/fvets.2020.573106

33. Tsang B, Gerlai R. Researchers, animal support and regulatory staff: symbiosis or antagonism? Laboratory Animal Research. 2022;38: 19. doi:10.1186/s42826-022-00129-0

34. Paull GC, Lee CJ, Tyler CR. Beyond compliance: harmonising research and husbandry practices to improve experimental reproducibility using fish models. Biological Reviews. 2023. doi: 10.1111/brv.13020

We look forward to receiving your response and decision. Thank you for inviting our revision. 

Sincerely, 

Megan LaFollette, MS, PhD and Co-Authors 

 

1. When submitting your revision, we need you to address these additional requirements. Please ensure that your manuscript meets PLOS ONE's style requirements, including those for file naming. The PLOS ONE style templates can be found at

We have revised the manuscript to ensure it meets all of PLOS ONE’s style requirements and have fixed all naming conventions.

“All authors are employed by the institution noted in their affiliation except for Lauren Young who is a student. All authors are also members of the 3Rs Collaborative’s Compassion Fatigue Initiative. Elizabeth Nunamaker and Sally Thompson-Iritani also sat on the board of directors and leadership team of the 3Rs Collaborative during publication.”

We have added the requested line to our Competing Interests statement in the above cover letter. Thank you for updating this on our behalf. We have also added this to our manuscript file.

This is still correct. We will provide repository information for our data, and there are no changes to our Data Availability statement.

4. Please remove your figures from within your manuscript file, leaving only the individual TIFF/EPS image files, uploaded separately. These will be automatically included in the reviewers’ PDF.

We have ensured the figures are removed from within the manuscript, leaving only the image files that were separately uploaded.

We have added captions for the Supporting Information files to submit along with our manuscript. 

We have added three new references to our reference list based on suggestions from Reviewer #2.

Reviewer #1: 

Dear Authors,

Congratulations on your work. It is an interesting manuscript that continues with your line of work and that contributes new data to a field that has more consideration.

Thank you to Reviewer 1 for these remarks.

1. Regarding statistics, I would propose to calculate the size effect, and make correlations to determine the influence of each variable studied.

We have now calculated effect sizes and added these to the Results section of our manuscript at lines 320-325. We briefly mention this in the discussion as well.

2. There is a typo in table 1, "veterinary technician" 910 instead of 10.

This is a great catch thank you. This correction to Table 1 has been made.

Reviewer #2: 

1. The manuscript by Young et al uses a mixed methods cross sectional survey to explore the quality of life of animal research personnel. Of particular interest is the finding that compassion fatigue is impacted most by institutional culture. On this point, it could be valuable for the authors to divide the respondents by level of seniority of the employee on the institutional culture metric.

This is a great suggestion. Based on our sample size per role, we only believed it was appropriate to further investigate our themes in animal caretakers and researchers. We do not have large enough sample sizes for veterinarians (n=10), managers (n=18) and the ‘other’ category (n=15), to even make explorative conclusions.

However, we did have a respectable sample size for animal caretakers and researchers. We divided them by the percentage of respondents in each role that mentioned each theme. We have added this to the Results at line 503-510 and have also added a new Figure 3 to display these findings. We have mentioned this in the discussion as well lines 608-612.

2. As mentioned by the authors, this study cannot account for those who have left their role. However, the inclusion of such data (for example, the turnover rate) of specific job categories, could be insightful. Whether their current institution is their 1st or 5th, and the reason for their departure from previous roles could also provide good info. From my experience, there is often a very high rate of departure and hiring in animal care roles.

Thank you for this suggestion. We are currently in the midst of updating our survey for another year of data collection, and this is something we will consider adding to our updated survey, to gather data on turnover rates for our longitudinal study.

3. Line 274 - Table 1 under veterinary technician typo? I believe the authors meant to put n of 10, not 910.

This is a great catch thank you. This correction to Table 1 has been made.

4. Papers of potential interest to the authors for this study.

https://www.nature.com/articles/s41684-021-00852-6

https://labanimres.biomedcentral.com/articles/10.1186/s42826-022-00129-0

https://www.frontiersin.org/articles/10.3389/fvets.2020.573106/full

https://onlinelibrary.wiley.com/doi/full/10.1111/brv.13020

Thank you for these references. We have read these publications and have included reference to three of them in our Discussion (lines 561-563 and lines 585-586) to further put our results in the context of the broader literature on compassion fatigue and a Culture of Care.

---

## [Decision Letter · Decision Letter 1]

26 Dec 2023

PONE-D-23-17551R1Professional quality of life in animal research personnel is linked to retention & job satisfaction: A mixed-methods cross-sectional survey on compassion fatigue in the USAPLOS ONE

Dear Dr. LaFollette,

Thank you for submitting your manuscript to PLOS ONE. After careful consideration, we feel that it has merit but does not fully meet PLOS ONE’s publication criteria as it currently stands. Therefore, we invite you to submit a revised version of the manuscript that addresses the points raised during the review process.

We look forward to receiving your revised manuscript.

Kind regards,

Rosemary Frey

Academic Editor

PLOS ONE

Journal Requirements:

Additional Editor Comments:

Please include the two articles requested the reviewer. 1

Reviewers' comments:

Reviewer's Responses to Questions

**Comments to the Author**

1. If the authors have adequately addressed your comments raised in a previous round of review and you feel that this manuscript is now acceptable for publication, you may indicate that here to bypass the “Comments to the Author” section, enter your conflict of interest statement in the “Confidential to Editor” section, and submit your "Accept" recommendation.

Reviewer #1: All comments have been addressed

Reviewer #2: All comments have been addressed

2. Is the manuscript technically sound, and do the data support the conclusions?

Reviewer #1: Yes

Reviewer #2: Yes

3. Has the statistical analysis been performed appropriately and rigorously? 

Reviewer #1: Yes

Reviewer #2: Yes

4. Have the authors made all data underlying the findings in their manuscript fully available?

Reviewer #1: Yes

Reviewer #2: Yes

5. Is the manuscript presented in an intelligible fashion and written in standard English?

Reviewer #1: Yes

Reviewer #2: Yes

6. Review Comments to the Author

Reviewer #1: Dear Authors,

Thank you for indicating the size effects. During this time I have noticed that two articles have been published in Laboratory Animals that are not mentioned in the bibliography and should be included. The first is a systematic review "Psychological stress and strain in laboratory animal professionals - a systematic review" (10.1177/00236772221129111) and the second is an article also looking at professional quality of life and mental health "Perceived professional quality of life and mental well-being among animal facility personnel in Spain" (10.1177/00236772231187177).

Best regards,

Reviewer #2: Congratulations on the work. This area of study is often overlooked, and I am positive both scientists and animal welfare personnel will benefit from the results of this study to improve their current working policies. Looking forward to reading the follow up studies in the future.

7. PLOS authors have the option to publish the peer review history of their article (what does this mean?). If published, this will include your full peer review and any attached files.

Reviewer #1: No

Reviewer #2: No

---

## [Author Response · Author response to Decision Letter 1]

17 Jan 2024

January 17, 2024 

Dear Dr. Rosemary Frey 

Thank you again for inviting us to submit a revised version of our manuscript and allowing us to make these additional revisions. We appreciate the additional feedback that the reviewers have provided. We have completed all revisions and address each comment individually below. The reviewer's comments are bolded, and our responses are non-bolded. 

We have also added the following two references to our reference list, based on helpful suggestions from Reviewer #1:

1. Rumpel, S., Kempen, R., Merle, R., & Thoene-Reineke, C. (2023). Psychological stress and strain in laboratory animal professionals – a systematic review. Laboratory Animals, 57(4), 396–411. https://doi.org/10.1177/00236772221129111

2. Chang, F. T., & Hard, L. A. (2002). Human-Animal Bonds in the Laboratory: How Animal Behavior Affects the Perspective of Caregivers. ILAR Journal, 43(1), 10–18. https://doi.org/10.1093/ilar.43.1.10

3. Friese, C., & Latimer, J. (2019). Entanglements in Health and Well-being: Working with Model Organisms in Biomedicine and Bioscience. Medical Anthropology Quarterly, 33(1), 120–137. https://doi.org/10.1111/maq.12489

4. Engel, R. M., Silver, C. C., Veeder, C. L., & Banks, R. E. (2020). Cognitive Dissonance in Laboratory Animal Medicine and Implications for Animal Welfare. Journal of the American Association for Laboratory Animal Science: JAALAS, 59(2), 132–138. https://doi.org/10.30802/AALAS-JAALAS-19-000073

5. Goñi-Balentziaga, O., & Azkona, G. (2023). Perceived professional quality of life and mental well-being among animal facility personnel in Spain. Laboratory Animals, 00236772231187177. https://doi.org/10.1177/00236772231187177

We look forward to receiving your response and decision. Thank you for inviting our revision. 

Sincerely, 

Megan LaFollette, MS, PhD and Co-Authors 

 

Journal Requirements:

We have added two additional references per Reviewer #1’s request. We have noted the additional citations in the revised manuscript and in the cover letter above.

Additional Editor Comments:

Please include the two articles requested the reviewer. 1

We have added the two additional articles per Reviewer #1’s request. 

Reviewers' comments:

Review Comments to the Author

Reviewer #1: Dear Authors,

Thank you for indicating the size effects. During this time I have noticed that two articles have been published in Laboratory Animals that are not mentioned in the bibliography and should be included. The first is a systematic review "Psychological stress and strain in laboratory animal professionals - a systematic review" (10.1177/00236772221129111) and the second is an article also looking at professional quality of life and mental health "Perceived professional quality of life and mental well-being among animal facility personnel in Spain" (10.1177/00236772231187177).

Best regards,

Thank you to Reviewer #1 for this suggestion to add these articles. These are very important findings to mention in comparison to our findings. We have integrated the first article into our Introduction at lines 77-90, as well as three articles that this systematic review references, and we have incorporated its findings into our Discussion at many different points throughout. We have also incorporated the second article into our Discussion at 579-580 and 606. These two articles have also been added to our Reference List, along with three additional articles that Rumpel et al. 2023 reference whose findings we found to be relevant.

Reviewer #2: Congratulations on the work. This area of study is often overlooked, and I am positive both scientists and animal welfare personnel will benefit from the results of this study to improve their current working policies. Looking forward to reading the follow up studies in the future.

Thank you to Reviewer #2 for the positive feedback on this work. We also look forward to continuing this work and conducting follow up surveys to build on our findings.

---

## [Decision Letter · Decision Letter 2]

30 Jan 2024

Professional quality of life in animal research personnel is linked to retention & job satisfaction: A mixed-methods cross-sectional survey on compassion fatigue in the USA

PONE-D-23-17551R2

Dear Dr.,LaFollette,

We’re pleased to inform you that your manuscript has been judged scientifically suitable for publication and will be formally accepted for publication once it meets all outstanding technical requirements.

Kind regards,

Rosemary Frey

Academic Editor

PLOS ONE

Additional Editor Comments (optional):

Reviewers' comments:

Reviewer's Responses to Questions

**Comments to the Author**

1. If the authors have adequately addressed your comments raised in a previous round of review and you feel that this manuscript is now acceptable for publication, you may indicate that here to bypass the “Comments to the Author” section, enter your conflict of interest statement in the “Confidential to Editor” section, and submit your "Accept" recommendation.

Reviewer #1: All comments have been addressed

Reviewer #2: All comments have been addressed

2. Is the manuscript technically sound, and do the data support the conclusions?

Reviewer #1: Yes

Reviewer #2: Yes

3. Has the statistical analysis been performed appropriately and rigorously? 

Reviewer #1: Yes

Reviewer #2: Yes

4. Have the authors made all data underlying the findings in their manuscript fully available?

Reviewer #1: Yes

Reviewer #2: Yes

5. Is the manuscript presented in an intelligible fashion and written in standard English?

Reviewer #1: Yes

Reviewer #2: Yes

6. Review Comments to the Author

Reviewer #1: Dear Authors

Thank you very much for updating the references. I have not found this one in the text: Goñi-Balentziaga, O., & Azkona, G. (2023). Perceived professional quality of life and mental well-being among animal facility staff in Spain. Animales de Laboratorio, 00236772231187177. https://doi.org/10.1177/00236772231187177

But otherwise, everything is fine.

Best regards,

Reviewer #2: The authors have addressed my previous comments. No further questions or revisions are requested from me.

7. PLOS authors have the option to publish the peer review history of their article (what does this mean?). If published, this will include your full peer review and any attached files.

Reviewer #1: No

Reviewer #2: No

---

## [Editor Report · Acceptance letter]

2 Apr 2024

PONE-D-23-17551R2 

PLOS ONE

Dear Dr. LaFollette, 

I'm pleased to inform you that your manuscript has been deemed suitable for publication in PLOS ONE. Congratulations! Your manuscript is now being handed over to our production team.

Kind regards, 

on behalf of

Dr. Rosemary Frey 

Academic Editor

PLOS ONE